# Latent-Guided Equivariant Diffusion for
# Controlled Structure-Based De Novo Ligand Generation

**Tuan Le** [* 1 2]  **Julian Cremer** [* 1 3]  **Djork-Arné Clevert** [1]  **Kristof Schütt** [1]

## Abstract

We propose PoLiGenX for *de novo* ligand design using latent-conditioned, target-aware equivariant diffusion. Our model leverages the conditioning of the generation process on reference molecules within a protein pocket to produce shape-similar *de novo* ligands that can be used for target-aware hit expansion and hit optimization. The results of our study showcase the efficacy of PoLiGenX in ligand design. Docking scores indicate that the generated ligands exhibit superior binding affinity compared to the reference molecule while preserving the shape. At the same time, our model maintains chemical diversity, ensuring the exploration of diverse chemical space. The evaluation of Lipinski's rule of five suggests that the sampled molecules possess a higher drug-likeness than the reference data. This constitutes an important step towards the controlled generation of therapeutically relevant *de novo* ligands tailored to specific protein targets.

## 1. Introduction

In recent years, the intersection of artificial intelligence (AI) and drug discovery has witnessed remarkable strides, with the potential to revolutionize the traditional approaches to identifying novel therapeutic compounds. Among these innovations, AI-enabled structure-based drug discovery has emerged as a promising research avenue, in particular in form of equivariant target-aware diffusion models. By conditioning the diffusion process on the receptors of proteins, these models exhibit a remarkable capacity to generate *de novo* ligands with enhanced affinity (Peng et al., 2022; Guan et al., 2023; Schneuing et al., 2023; Le et al., 2024). Failing to consider the essential chemical properties for target binding can lead to a significant lack of specificity and result in ineffective drug candidates. Moreover, these candidates must exhibit favorable absorption, distribution, metabolism, excretion (ADME), and toxicity profiles. Designing ligands from scratch without addressing these critical properties may produce molecules with poor bioavailability or potential toxicity, thereby limiting their therapeutic potential. This challenge is further exacerbated by the often sparse and noisy data available for developing effective machine learning models. However, machine learning shows considerable promise during the hit expansion phase of drug discovery. This crucial stage involves enhancing and exploring the chemical space around promising hits already identified through high-throughput screening or other methods. In this study, we introduce PoLiGenX (**Po**cket-based **Li**gand **Gen**erator for hit e**X**pansion) that generates ligands *de novo* within a protein binding pocket. Unlike previous models, PoLiGenX starts with a seed molecule, such as a hit candidate or an initial scaffold, and iteratively refines and modifies it to improve its efficacy. We enhance the capabilities of the existing equivariant diffusion model, EQGAT-diff (Le et al., 2024), by incorporating a latent encoding as a condition. It is derived from an invariant graph neural network that is jointly trained to process 3D molecular inputs. The setup ensures that the newly generated ligands retain structural characteristics of the seed molecules while undergoing necessary chemical modifications and diversification. Our proposed approach adds a new level of control to the process of generating *de novo* ligands, aligning it more closely with the specific needs of targeted drug design, particularly during the hit expansion phase.

## 2. Related Work

Deep generative modeling in the life sciences has become a promising research area. Recent work by Xu et al. (2022); Jing et al. (2022) uses Denoising Diffusion Probabilistic Models (DDPMs) (Sohl-Dickstein et al., 2015; Ho et al., 2020; Kingma et al., 2021; Song et al., 2021) to predict the 3d coordinates of molecules with the help of 3d equivariant graph neural networks. In the *de novo* setting, another line of research focuses on directly generating the atomic coordinates and elements, using autoregressive models (Gebauer

---

[*]Equal contribution [1]Pfizer Research & Development [2]Freie Universität Berlin [3]University Pompeu Fabra. Correspondence to: Tuan Le <tuan.le@pfizer.com>, Julian Cremer <julian.cremer@pfizer.com>.

*Accepted at the 1st Machine Learning for Life and Material Sciences Workshop at ICML 2024.* Copyright 2024 by the author(s).

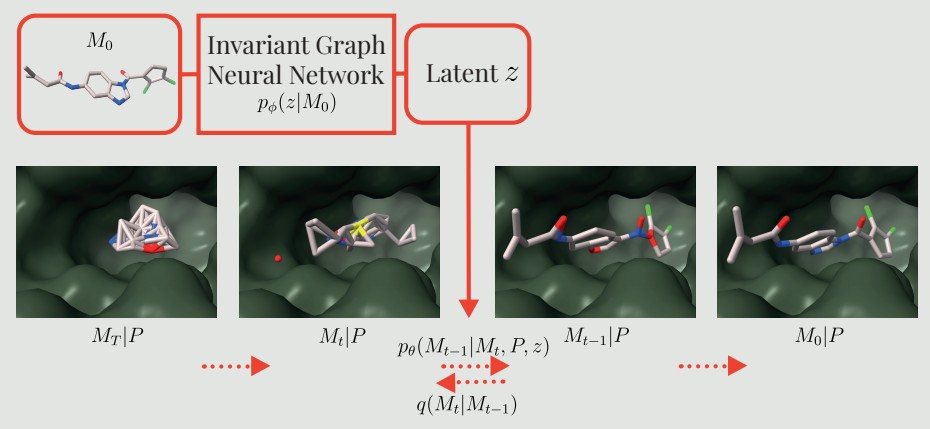

*Figure 1.* Graphical model of the proposed latent diffusion model. The encoded ligand $z$ serves as input to the diffusion model $p_\theta$ to steer the generation process of new ligands $M_0$.

et al., 2019; 2022; Luo & Ji, 2022). Hoogeboom et al. (2022) introduced E(3) equivariant diffusion model (EDM) for *de novo* molecule design that simultaneously learns atomic elements next to the coordinates while treating chemical elements as continuous variables to utilize the formalism of DDPM. Follow-up work leverages EDM and develop diffusion models for linker design (Igashov et al., 2022) and structure-based ligand modeling (Schneuing et al., 2023; Guan et al., 2023; Le et al., 2024). In the context of shape-conditioned molecule generation Adams & Coley (2023) (SQUID) and Chen et al. (2023) (ShapeMol) recently proposed to incorporate the shape of a seed molecule into the generation process. Both approaches use an equivariant surface encoding of a seed molecule, whereby SQUID uses variational auto-encoding on graphs and focuses more on fragment-based design. ShapeMol is an adaption of SQUID in 3d space leveraging an equivariant diffusion model. However, both works do not include a protein receptor condition. We propose to use a simple approach employing reference molecules in a latent representation, as outlined in more details below.

## 3. Methods

**Problem Formulation and Notation**   We investigate the generation of molecular structures $M$ in a *de novo* setting conditioned on a protein pocket $P$, i.e., building a generative model $p_\theta(M|P)$. For this, we use the EQGAT-diff framework proposed by Le et al. (2024). In this setup, a noisy ligand $M_t = (X_t, H_t, E_t)$ — representing perturbed atomic coordinates, element types, and bond features — is used, and the diffusion model $p_\theta$ predicts the uncorrupted data modalities $(\hat{X}_0, \hat{H}_0, \hat{E}_0)$, because the distribution $M_{t-1}|M_t$ depends on both $M_t$ and $\hat{M}_0$. Specifically, for continuous coordinates, the reverse distribution adheres to a multivariate Gaussian model, while for discrete-valued modalities,

it follows a categorical distribution. We refer to Le et al. (2024) for further details.

While models like EQGAT-diff, TargetDiff or DiffSBDD generate ligands in context of a protein pocket, they do not constraint the generated ligands to preserve properties like shape or chemical similarity during training. In contrast, we include a latent variable $z \in \mathbb{R}^K$ that relates to the input molecule $\hat{M}_0$. The latent $z$ may serve as a shape conditioning that also comprises chemical information like the atom composition of the molecule $\hat{M}_0$. The graph encoder $q_\phi : \mathcal{X}^M \to \mathbb{R}^K$ is invariant to permutation, rotation and translation of atoms (Winter et al., 2022; Le et al., 2022).

Following Adams & Coley (2023), chemical similarity of two molecules is measured as the Tanimoto similarity of ECFP4 fingerprints (2048 bits) computed by RDKit, whereby shape similarity is defined by Gaussian descriptions of molecular shape in form of atom-centered Gaussians and calculated by the volume overlaps between them as in Adams & Coley (2023).

**PoLiGenX**   To model the dependence on variable $z$, we include a variational distribution $q_\phi(z|M_0)$ similar to Luo & Hu (2021); Zeng et al. (2022) and obtain the ELBO

$$p_\theta(M_0|P) = \mathbb{E}_{q(M_{1:T}|M_0)q_\phi(z|M_0)}\Big[\frac{p_\theta(M_0, M_{1:T}, z|P)}{q(M_{1:T}|M_0)q_\phi(z|M_0)}\Big]$$

$$\geq \mathbb{E}_{q(M_{1:T}|M_0)q_\phi(z|M_0)}\Big[\log \frac{p_\theta(M_0, M_{1:T}, z|P)}{q(M_{1:T}|M_0)q_\phi(z|M_0)}\Big]$$

$$= \mathbb{E}_{q(M_1|M_0)q_\phi(z|M_0)}[\log p_\theta(M_0|M_1, P, z)]$$

$$+ \mathbb{E}_{q(M_T|M_0)q_\phi(z|M_0)}\Big[\log \frac{p(M_T|z)}{q(M_T|M_0)}\Big]$$

$$- D_{KL}(q_\phi(z|M_0)||p(z)) - \sum_{t=2}^{T} \mathbb{E}_{q(M_t|M_0)q_\phi(z|M_0)}[L_{t-1}],$$

$$\tag{1}$$

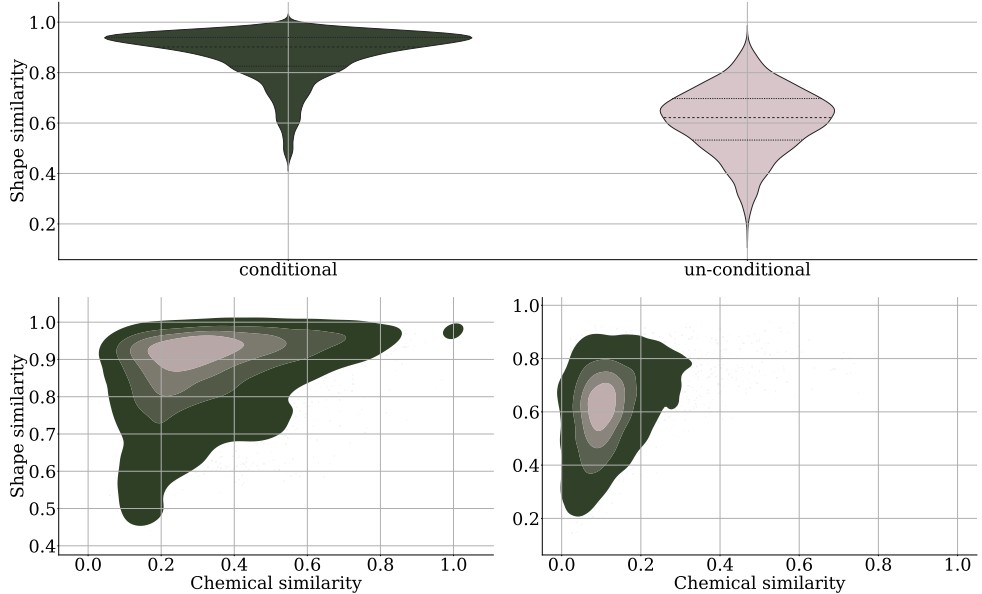

*Figure 2.* **Top**: Violin plot of the Tanimoto shape similarity evaluated across all test targets of the CrossDocked dataset. PoLiGenX (left) is compared to EQGAT-diff (right). In the conditional setting the model generates significantly more shape-similar molecules. **Bottom**: Heatmap histogram comparing PoLiGenX (left) with EQGAT-diff with respect to Tanimoto shape and chemical similarity on the CrossDocked test set. The brighter the color the higher the molecule count.

where the diffusion loss $L_{t-1}$ is per timestep and defined as $L_{t-1} = D_{KL}(q(M_{t-1}|M_t, M_0)||p_\theta(M_{t-1}|M_t, P, z))$.

We extend the diffusion model by a conditioning on $z$ and train $p_\theta(M|P, z)$ to minimize the KL divergence to the tractable reverse distribution, which is achieved when predicting the original data points $\hat{M}_0$ (Ho et al., 2020; Austin et al., 2021; Le et al., 2024). Similar to prior works, we optimize the diffusion $L_{t-1}$ by drawing steps per minibatch instead of the entire trajectory. We adopt a Gaussian prior for the latent distribution, i.e., $p(z) \sim N(0, I)$ and enforce a smooth latent space by choosing the maximum mean discrepancy (MMD) loss (Tolstikhin et al., 2018) over the KL divergence. The prior distribution for the ambient data space, i.e., $M_T$ is a 0-CoM Gaussian for coordinates and empirical categorical distribution for discrete data types from the training set as discussed in Le et al. (2024). During training, we sample a batch of pocket-ligand pairs and a step $t \in \{1, \dots, 500\}$. Next, we encode the ligands $M_0$ into latents $z$, apply the noise process to the ligands to obtain $M_t$ and minimize the diffusion loss while providing $z$ as an additional input via adaptive layer normalization (Huang & Belongie, 2017) next to the protein pocket $P$. We refer to the supplementary material for further details including the derivation of the ELBO.

## 4. Results

We train PoLiGenX using the CrossDocked2020 (Francoeur et al., 2020) dataset, following the same dataset splits as found in previous research (Luo et al., 2021; Peng et al., 2022; Guan et al., 2023; Schneuing et al., 2023; Le et al., 2024). Unlike other models, PoLiGenX incorporates not only the protein pocket as a condition for generating novel ligands but also utilizes a latent embedding of a ligand from the dataset as an initial condition. This distinctive approach positions PoLiGenX differently from the mentioned models — it is specifically designed to perform tasks akin to hit expansion by enhancing specificity, chemical diversity, and binding affinity, rather than operating solely as a target-aware, but unconditional *de novo* model. In the following, we evaluate if PoLiGenX effectively maintains the structural shape of the seed molecule while promoting chemical diversity.

Fig. 2 (top) shows the evaluation of the mean shape similarity on the CrossDocked test set for both PoLiGenX (conditional) and EQGAT-diff (unconditional). The test set comprises 100 ligand-pocket complexes for which 100 ligands each were sampled and the Tanimoto shape similarity measured against the reference ligands. PoLiGenX exhibits significantly higher shape similarities across complexes. However, we aim to preserve the shape between reference and sample *without* sacrificing chemical diversity to ensure an efficient exploration of chemical space. Fig. 2 (bottom) shows the distribution of shape similarity against

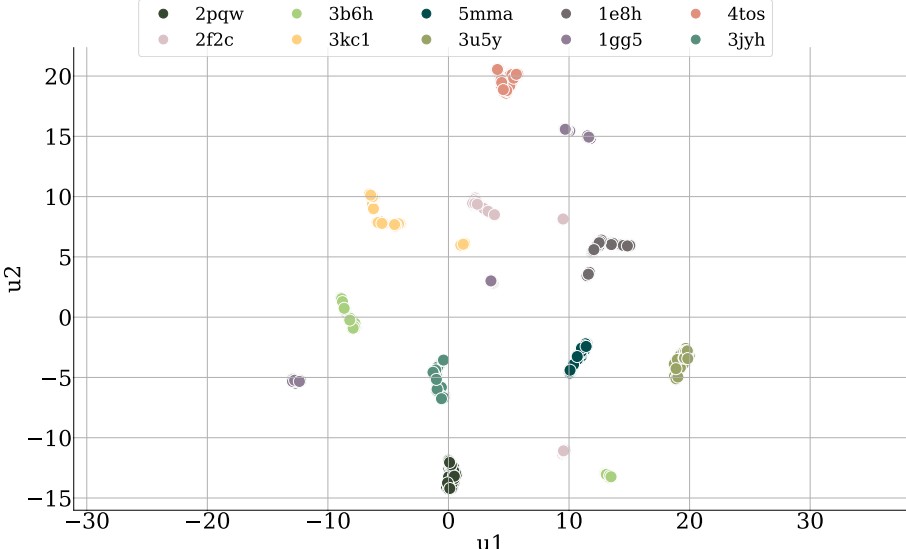

Figure 3. UMAP plot showing the 2d projections of the latent embeddings of 100 sampled ligands per target for ten randomly sampled test set targets.

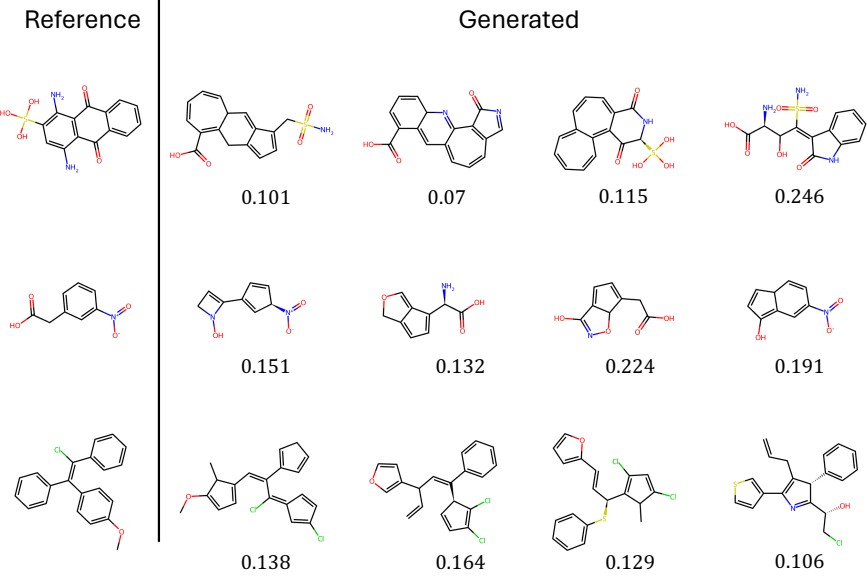

Figure 4. Reference molecules extracted from the CrossDocked test split (left) and four generated molecules sampled randomly with PoLiGenX. Below each generated ligand, we also show the chemical similarity to the reference ligand.

chemical similarity for conditional and unconditional sampling. We observe a mean shape similarity of $0.64$ and $0.12$ chemical similarity for EQGAT-diff. In contrast, PoLiGenX exhibits a significant increase in shape similarity with mean value of $0.87$, but also generates a reasonably high diversity in samples with mean chemical similarity of $0.33$.

To evaluate the expressiveness of the learned latent embeddings, Fig. 3 visualizes the UMAP projections of the latent embeddings. We sampled 100 ligands per receptor for ten

randomly selected targets of the CrossDocked test set. The resulting UMAP projections reveal that the latent embeddings effectively separate the ligands into distinct clusters specific to each target. This observation suggests that our latent model successfully captures the context of ligands in relation to their respective protein receptors.

Next, we compare molecules sampled conditionally from our model, PoLiGenX, with the reference test data, focusing on docking scores and chemical properties. As previously

*Table 1.* Docking performance on the CrossDocked test set and ligands generated using PoLiGenX. QuickVina2 is employed for docking. We report mean values across all targets with standard deviations given as subscripts. Drug-likeness is measured via RDKit's QED value. Further, molecules are evaluated in terms of the octanol–water partition coefficient (logP), the molecular weight (MolWt) and the number of hydrogen acceptors and donors. Following Lipinski's rule of five, we report the percentage of molecules that obey the respective rule. The last column gives the average of molecules fulfilling all rules.

| Data | QVina2 (All) ↓ | QVina2 (Top-10%) ↓ | QED ↑ | logP ↑ | MolWt ↑ | H-acceptors ↑ | H-donors ↑ | Lipinski ↑ |
|------|---------------|--------------------|-------|--------|---------|---------------|------------|-----------|
| CrossDocked test set | $-6.85_{\pm 2.33}$ | - | $0.47_{\pm 0.20}$ | 0.79 | 0.85 | 0.84 | 0.8 | $3.35_{\pm 1.14}$ |
| PoLiGenX | $\mathbf{-7.21}_{\pm 2.22}$ | $\mathbf{-8.04}_{\pm 2.44}$ | $\mathbf{0.59}_{\pm 0.20}$ | **0.91** | **0.87** | **0.85** | **0.91** | $\mathbf{3.57}_{\pm 0.93}$ |

outlined, the purpose of PoLiGenX is significantly different to recent *de novo* models, such as EQGAT-diff, hence we omit a comparison. Tab. 1 summarizes the results. We observe improved docking scores for generated samples compared to the CrossDocked test data, in particular within the top 10% of each target. Here, we reach a docking score of $-8.04 \pm 2.44$ compared to $-6.85 \pm 2.33$ for the test data. At the same time, the generated ligands per target show improvement in RDKit's drug-likeness score (QED) and adherence to Lipinski's Rule of Five. These are chemical features recognized from a medicinal chemistry perspective as guidelines to identify compounds likely to possess favorable bioavailability. Specifically, the octanol-water partition coefficient (logP) should be less than 5, molecular weight (MolWt) should be less than 500 Daltons, hydrogen bond acceptors (H-acceptors) less than 10 and hydrogen bond donors (H-donors) should be less than 5.

Fig. 4 depicts three randomly chosen test set ligands with four conditionally sampled and randomly selected ligands each. Judging by visual inspection, the topology is well preserved. We note that chemical similarity, especially based on fingerprints can change drastically if some chemical elements are interchanged. As shown in the bottom panel in Figure 2, PoLiGenX achieves a mean chemical similarity of around $0.33$ while preserving shape similarity of $0.87$ compared to the unconditional case with $0.12$ and $0.64$ for chemical and shape similarity, respectively.

The controlable generation of PoLiGenX can be further regulated by including a control parameter $\lambda \in (0, 1]$ that scales the latent $z$ when going into the diffusion model. That is, for small $\lambda$ values approaching 0, PoLiGenX does not include any latent information and collapses to the unconditional EQGAT-diff and only leverages the pocket information as context. With $\lambda$ interpolating between e.g. $(0.5, 1.0)$, we observe that the mean chemical similarity for generated ligands with respect to the references also increases as depicted in Figure 5. We detail the influence of the latent variable $z$ in combination with the scale parameter $\lambda$ in the supplementary materials.

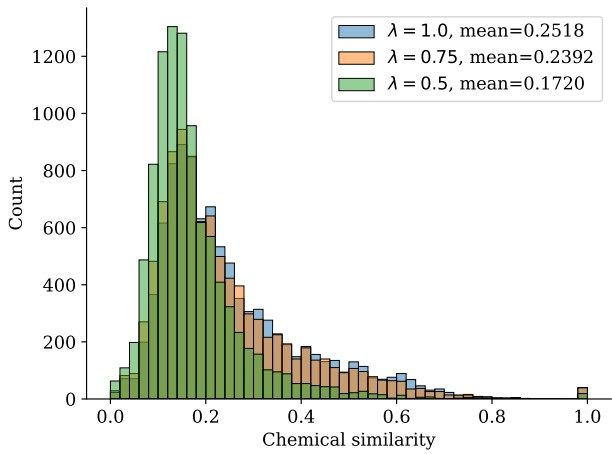

*Figure 5.* Density plot for chemical similarity of generated ligands from PoLiGenX with varying $\lambda$ control parameter. With increasing $\lambda$, the latent $z$ of reference/seed ligand $M_0$ is preserved such that generated ligands exhibit higher chemical similarity to $M_0$.

## 5. Conclusions

We have developed PoLiGenX for controlled *de novo* ligand generation within a protein binding pocket. By incorporating a latent encoding from a seed molecule into the diffusion model, we ensure that the generated ligands preserve shape and also adhere to the structural constraints of the target protein binding site. The effectiveness of PoLiGenX is evidenced by improved docking scores compared to reference ligands. Additionally, the generated ligands conform to Lipinski's Rule of Five, demonstrating their drug-likeness. Importantly, the model maintains chemical diversity, which is essential for exploring a broad range of chemical space and discovering novel therapeutic candidates. This integration of shape preservation, target specificity, and chemical diversity provides a powerful approach for the targeted generation of drug candidates, particularly useful in the hit expansion phase of drug discovery campaigns.

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
