# Supplementary Material: Latent-Guided Equivariant Diffusion for Controlled Structure-Based De Novo Ligand Generation

**Tuan Le** [* 1 2]  **Julian Cremer** [* 1 3]  **Djork-Arné Clevert** [1]  **Kristof Schütt** [1]

This document encloses additional information related to the submitted workshop paper *Latent-Conditioned Equivariant Diffusion for Structure-Based De Novo Ligand Generation*.

## A. Model and Training Details

We leverage the EQGAT-diff architecture as proposed in Le et al. (2024) which and modify it to process the pocket-ligand (PL) complex. To build the PL complex, we use the creation strategy from DiffSBDD Schneuing et al. (2023) with a 5 Ångström cutoff. In summary, a PL complex is built by iterating over the atoms in each residues in a protein and computing all pairwise distances to all ligand atoms. If any distance between the residues' atom to any ligand atom is below the cutoff, the entire residue is included in the pocket.

To form the PL representation for message-passing we stack ligand and pocket features. Ligands represented as $(X_l, H_l, E_l)$, indicating spatial coordinates, atom and edge types are in same fashion generated as in EQGAT-diff (Le et al., 2024). That is, we create a fully-connected graph and include the bond adjacency with features (single, double, triple and no bond) into the edge features.

The edge features in the pocket representation $(X_p, H_p, E_p)$, are all set to the no bond type. The edge indices and connectivity for pocket-pocket and ligand-pocket/pocket-ligand interaction are obtained through a radius graph with 5A cutoff.

An important aspect for training diffusion models is the noise scheduler, particularly when different modalities such as coordinates, atom- and bond-types are learnt. In similar fashion to the work by (Vignac et al., 2023), we leverage an adaptive noise scheduler

$$\bar{\alpha}^t = \cos\left(\frac{\pi}{2}\frac{(t/T + s)^\nu}{1+s}\right)^2,$$

with $\nu_r = 2.5, \nu_y = 1.5, \nu_x = 1.0$ denoting for atom coordinates, bond types, atom types, respectively. The rationale behind these coefficients is to enable for slower decay of the signals of coordinates, bond- and atom-types.

We provide a high-level computational workflow in Figure 1.

### A.1. G-invariant latent encoder EQGAT

The group-invariant graph encoder is implemented using the EQGAT message passing layer as proposed in Le et al. (2022) whose computational graph is visualized in their Figure 1b. After $L = 8$ round of message passing, we extract the SO(3)-invariant scalar node embeddings $H \in \mathbb{R}^{N \times k}$ and leverage a gated equivariant transformation as proposed in the original EQGAT architecture (Le et al., 2022) followed by a SoftmaxAttention-pooling (Li et al., 2019) along the node embeddings, to achieve the SO(3) as well as permutation invariant latent embedding $z \in \mathbb{R}^k$. We set the latent dimension to $k = 128$.

The derivation of the ELBO in B requires the variational distribution $z \sim q(z|x_0)$. Such distribution can be simply obtained through e.g., reparameterization using Gaussian variables as $z = \mu_\phi(x_0) + \sigma_\phi(x_0) \odot \epsilon,\ \epsilon \sim N(0, I_K)$. Here, we follow

---

[*]Equal contribution [1]Pfizer Research & Development [2]Freie Universität Berlin [3]University Pompeu Fabra. Correspondence to: Tuan Le <tuan.le@pfizer.com>, Julian Cremer <julian.cremer@pfizer.com>.

*Accepted at the 1st Machine Learning for Life and Material Sciences Workshop at ICML 2024.* Copyright 2024 by the author(s).

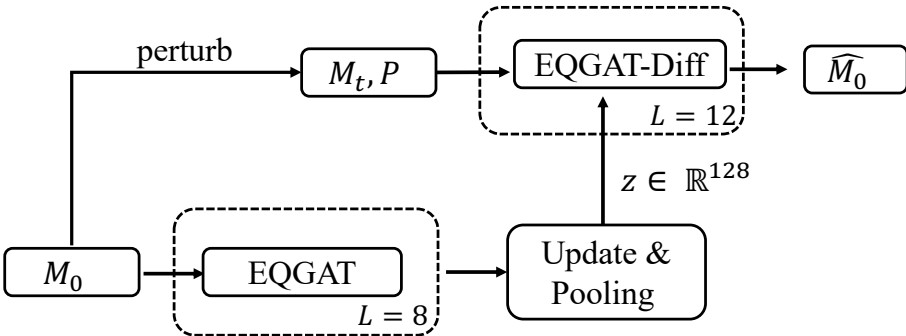

*Figure 1.* High level overview of the model computation. The lower part describes the group invariant graph encoder that inputs the ground-truth ligand $M_0$ and outputs the latent embedding $z$. EQGAT-diff inputs the perturbed ligand $M_t$ next to the pocket as well as latent representation as context. EQGAT-diff is tasked in predicting the uncorrupted ligand $M_0$.

the approach of deterministic (V)AE where the standard deviation converges towards 0, i.e. each latent being a dirac-delta point mass in $\mathbb{R}^K$.

## A.2. Incorporating the latent $z$ into EQGAT-diff

We include the latent embedding $z \in \mathbb{R}^K$ derived from the group-invariant graph encoder, via adaptive layer normalization (AdaLN) to fuse the global latent embedding with the node embeddings $S \in \mathbb{R}^{N \times K}$ from the current point cloud that EQGAT-diff performs after every message-passing layer. That is, instead of having shared learnable affine parameters for every input, we compute the affine parameters based on the latent (style) $z$, which is inspired by adaptive instance normalization introduced by Huang & Belongie (2017).

$$\text{AdaptiveLN}(H, z) = s_\theta(z) \odot \left( \frac{H - \mu(H)}{\sigma(H)} \right) + b_\theta(z), \tag{1}$$

where $\mu, \sigma$ are functions that compute the mean and standard deviation embeddings $\in \mathbb{R}^K$ from the hidden node embeddings $H \in \mathbb{R}^{N \times K}$. The influence of the latent embedding is enforced through the scale and shift operation obtained through the transformations $s_\theta, b_\theta$ which are both simple linear layers.

## A.3. Training

We train PoLiGenX under the data parameterization leveraging Gaussian diffusion for coordinates and categorical diffusion for discrete-valued data modalities, including chemical elements and bond types. Therefore, the loss function for a sampled timestep on the diffusion loss reads

$$L_{t-1} = w_s(t) \Big( \lambda_x ||X_0 - \hat{X}_0||^2 + \lambda_h \text{CE}(H_0, \hat{H}_0) + \lambda_e \text{CE}(E_0, \hat{E}_0) \Big), \tag{2}$$

where CE refers to the cross-entropy loss and $(\lambda_x, \lambda_h, \lambda_e) = (3, 0.4, 2)$ are weighting coefficients adapted from (Vignac et al., 2023). The loss weighting $w_s(t)$ is modified truncated signal-to-noise ratio in Le et al. (2024).

The EQGAT-diff model uses 256 scalar and vector features each and 128 edge features across 12 layers of fully connected message passing. This corresponds to 13.4M trainable parameters.

The EQGAT graph-encoder model uses 128 scalar and vector features and 16 edge features across 8 layers of local message passing based on a 5A radius cutoff. The graph-encoder has 2.0M trainable parameters. We compute the MMD loss between a batch of latents $Z$ with a random batch drawn from an isotropic Gaussian to enforce a smooth latent space as proposed by (Tolstikhin et al., 2018). Apart from that, we implement a simple (self)-supervised learning task in that the graph-encoder also predicts the number of nodes of the input molecule. This objective is optimized using cross-entropy loss, while additional self-supervised learning task might also be suitable, e.g., regressing on chemical properties that can be easily computed by the RDKit.

To train PoLiGenX, we use 8 NVIDIA A100 32GB GPUs with batch-size 8 for 300 epochs on the CrossDocked2020 5A

dataset similar to (Schneuing et al., 2023). As optimizer we use AdamW with learning rate $2 \cdot 10^{-4}$, weight-decay of $1 \cdot 10^{-12}$, and gradient clipping for values higher than 10.

## B. Derivation ELBO

We derive the variational lower bound for the latent diffusion model acting on data $x_0 \in \mathbb{R}^D$, where the latent variable is lower-dimensional $z \in \mathbb{R}^K$ with $K < D$. Since we leverage diffusion models, which construct a sequence of latent variables of the same size, i.e., $x_1, x_2, ..., x_T$ with $x_i \in \mathbb{R}^D$, the following general equation holds through marginalization

$$p(x_0) = \int p(x_0, x_1, \ldots, x_T) dx_1 dx_2, \ldots dx_T \tag{3}$$

$$= \int p(x_0, x_{1:T}) dx_{1:T}. \tag{4}$$

We use $p(x_{0:T})$ as abbreviation for $p(x_0, x_1, \ldots x_T)$ and include two variational distributions (VD) $q(z|x_0)$ as well as $q(x_{1:T}|x_0)$. The first VD $q(z|x_0)$ is the commonly known from the VAE (Kingma & Welling, 2014), while the second $q(x_{1:T}|x_0)$ is the Markov model (Sohl-Dickstein et al., 2015) which factorizes as

$$q(x_{1:T}|x_0) = \prod_{t=1}^{T} q(x_t|x_{t-1}), \tag{5}$$

by the Markov assumption $(*)$ where the next state only depends on the previous. Note that this also states for the reverse. We develop the ELBO as

$$
\begin{aligned}
p(x_0) &= \int p(x_{0:T}, z) dx_{1:T} dz \\
&= \int \frac{p(x_{0:T}, z) q(x_{1:T}|x_0) q(z|x_0)}{q(x_{1:T}|x_0) q(z|x_0)} dx_{1:T} dz = \mathbb{E}_{q(x_{1:T}|x_0)q(z|x_0)} \left[ \frac{p(x_{0:T}|z)p(z)}{q(x_{1:T}|x_0)q(z|x_0)} \right] \\
&\overset{*}{=} \mathbb{E}_{q(x_{1:T}|x_0)q(z|x_0)} \left[ \frac{p(x_T|z)p(z) \prod_{t=1}^{T} p(x_{t-1}|x_t, z)}{q(z|x_0) \prod_{t=1}^{T} q(x_t|x_{t-1})} \right] \\
&\geq \mathbb{E}_{q(x_{1:T}|x_0)q(z|x_0)} \left[ \log \left( \frac{p(x_T|z)p(z) \prod_{t=1}^{T} p(x_{t-1}|x_t, z)}{q(z|x_0) \prod_{t=1}^{T} q(x_t|x_{t-1})} \right) \right] \\
&= \mathbb{E}_{q(x_{1:T}|x_0)q(z|x_0)} \left[ \log \left( \frac{p(x_T|z)p(z)p(x_0|x_1, z)}{q(z|x_0)q(x_1|x_0)} \prod_{t=2}^{T} \frac{p(x_{t-1}|x_t, z)}{q(x_t|x_{t-1})} \right) \right],
\end{aligned}
\tag{6}
$$

where we condition on $x_0$ for the forward variatonal distribution $q(x_t|x_{t-1}) = q(x_t|x_{t-1}, x_0)$ because we require it later to obtain closed-form solution to regress on. We have as a side computation (using Bayes theorem)

$$
\begin{aligned}
\prod_{t=2}^{T} q(x_t|x_{t-1}, x_0) &= \prod_{t=2}^{T} \frac{q(x_{t-1}|x_t, x_0) q(x_t|x_0)}{q(x_{t-1}|x_0)} = \prod_{t=2}^{T} q(x_{t-1}|x_t, x_0) \prod_{t=2}^{T} \frac{q(x_t|x_0)}{q(x_{t-1}|x_0)} \\
&= \prod_{t=2}^{T} q(x_{t-1}|x_t, x_0) \cdot \frac{q(x_T|x_0)}{q(x_1|x_0)},
\end{aligned}
\tag{7}
$$

where $\prod_{t=2}^{T} \frac{q(x_t|x_0)}{q(x_{t-1}|x_0)}$ is a telescoping series cancelling out intermediate products. We can insert (7) into the product series in (6) to obtain

$$
\begin{aligned}
p(x_0) &\geq \mathbb{E}_{q(x_{1:T}|x_0)q(z|x_0)}\left[\log\left(\frac{p(x_T|z)p(z)p(x_0|x_1,z)}{q(z|x_0)q(x_1|x_0)}\frac{q(x_1|x_0)}{q(x_T|x_0)}\prod_{t=2}^{T}\frac{p(x_{t-1}|x_t,z)}{q(x_{t-1}|x_t,x_0)}\right)\right] \\
&= \mathbb{E}_{q(x_{1:T}|x_0)q(z|x_0)}\left[\log\left(\frac{p(x_T|z)p(z)p(x_0|x_1,z)}{q(z|x_0)q(x_T|x_0)}\prod_{t=2}^{T}\frac{p(x_{t-1}|x_t,z)}{q(x_{t-1}|x_t,x_0)}\right)\right] \\
&= \mathbb{E}_{q(x_{1:T}|x_0)q(z|x_0)}\left[\log p(x_0|x_1,z) + \log\frac{p(z)}{q(z|x_0)} + \log\frac{p(x_T|z)}{p(x_T|x_0)} + \sum_{t=2}^{T}\log\frac{p(x_{t-1}|x_t,z)}{q(x_{t-1}|x_t,x_0)}\right].
\end{aligned}
\tag{8}
$$

Next we evaluate the expectation, which is the (multi-dimensional) integral over the variational distributions $q(x_{1:T}|x_0)q(z|x_0)$, i.e., integrating over the domains of $x_{1:T}$ and $z$, to obtain

$$
\begin{aligned}
p(x_0) \geq &\mathbb{E}_{q(x_1|x_0)q(z|x_0)}[\log p(x_0|x_1,z)] + \mathbb{E}_{q(z|x_0)}[\log\frac{p(z)}{q(z|x_0)}] + \mathbb{E}_{q(x_T|x_0)q(z|x_0)}[\log\frac{p(x_T|z)}{q(x_T|x_0)}] \\
&+ \mathbb{E}_{q(x_{t-1}|x_t,x_0)q(x_t|x_0)q(z|x_0)}[\sum_{t=2}^{T}\log\frac{p(x_{t-1}|x_t,z)}{q(x_{t-1}|x_t,x_0)}],
\end{aligned}
\tag{9}
$$

where the (reverse) KL divergence between any distribution $q(x)$ and $p(x)$ reads

$$
D_{KL}(q(x)||p(x)) = \mathbb{E}_{q(x)}[\log\frac{p(x)}{q(x)}].
\tag{10}
$$

Plugging Eq. (10) into (8), we obtain

$$
p(x_0) \geq \mathbb{E}_{q(x_1|x_0)q(z|x_0)}[\log p(x_0|x_1,z)] + D_{KL}(p(z)||q(z|x_0)) + \mathbb{E}_{q(x_T|x_0)q(z|x_0)}[\log\frac{p(x_T|z)}{q(x_T|x_0)}]
\tag{11}
$$

$$
+ \mathbb{E}_{q(x_t|x_0)q(z|x_0)}[\sum_{t=2}^{T}D_{KL}(p(x_{t-1}|x_t,z)||q(x_{t-1}|x_t,x_0))].
\tag{12}
$$

The first expectation in the ELBO is commonly known as the reconstruction loss, while the second expectation is defined as KL-prior regularization loss. For the third expectation, we make the assumption that the ambient space prior $x_T$ is independet of the latent variable $z$, i.e., $p(x_T|z) = p(x_T)$. It could be interesting to further investigate if the ambient prior can be influenced through a latent variable but due to simplicity, we remove it. This leads to simplyfing the third expectation into $D_{KL}(p(x_T)||q(x_T|x_0)$, where by design of the diffusion model and its noise-schedule, $q(x_T|x_0)$ should converge to a multivariate standard normal distribution $N(0,I)$. The last expectation is the summation over $(T-1)$ KL-divergences to train the generative diffusion model matching the reverse $p(x_{t-1}|x_t,z)$ with the tractable $q(x_{t-1}|x_t,x_0)$. Instead of summing over all noise-levels indexed from $t=2$ until $t=T$ optimizing the derived ELBO, we only optimize $D_{KL}(p(z)||q(z|x_0))$ using the Maximum-Mean-Discrepancy (MMD) (Tolstikhin et al., 2018) loss instead of KL and uniformly sample a timestep from $t \sim U(2,T)$ to minimize $D_{KL}(p(x_{t-1}|x_t,z)||q(x_{t-1}|x_t,x_0))$.