# OpenReview forum: "Latent-Guided Equivariant Diffusion for Controlled Structure-Based De Novo Ligand Generation"
_ICML.cc/2024/Workshop/ML4LMS — ML4LMS Poster_

### Official Review · Reviewer_5ufm · 2024-06-05
**Interesting concept, limited validation**

**Rating:** 6
**Confidence:** 4

**Review:**

Pros:
Interesting concept, good referencing of recent SOTA.


Cons:
- The authors choose not to compare outcomes of their generated compounds with those of recent SOTA, I don't think this is justified
- The model they propose focuses on maintaining similar shape to the original seed compound, but there is limited justification for why this is important. Why shape and not some other calculated property?
- Figure 2 lacks calculation of statistical significance to support the author's claims
- Methods lacking e.g. dataset (compounds and receptors) source/processing

---

### Official Review · Reviewer_7fFB · 2024-06-12
**good paper, but in my opinion more comparisons could have been done to methods out there**

**Rating:** 6
**Confidence:** 3

**Review:**

Although PoligenX is distinct in its approach, I still think you can compare the results from docking performance to other docking methods than just QuickVina2. Maybe also a section comparing scores on binding affinity in PDBbind v2020 like in targetDiff for example.

Overall a good paper, that could do with more comparisons to other methods within different metrics like binding affinity, docking scores.

---

### Official Review · Reviewer_otkR · 2024-06-12
**Latent DDPM for constrained ligand generation**

**Rating:** 5
**Confidence:** 4

**Review:**

Decision: Marginally reject.

Summary: Latent DDPM for constrained ligand generation.

Remarks: I'd recommend putting the work through PoseCheck (https://arxiv.org/abs/2308.07413) to see if the generated molecules are valid within the protein pocket.